# Global metabolic rewiring for improved $CO_2$ fixation and chemical production in cyanobacteria

Masahiro Kanno[1,2], Austin L. Carroll[1] & Shota Atsumi[1]

Cyanobacteria have attracted much attention as hosts to recycle $CO_2$ into valuable chemicals. Although cyanobacteria have been engineered to produce various compounds, production efficiencies are too low for commercialization. Here we engineer the carbon metabolism of *Synechococcus elongatus* PCC 7942 to improve glucose utilization, enhance $CO_2$ fixation and increase chemical production. We introduce modifications in glycolytic pathways and the Calvin Benson cycle to increase carbon flux and redirect it towards carbon fixation. The engineered strain efficiently uses both $CO_2$ and glucose, and produces $12.6\,g\,l^{-1}$ of 2,3-butanediol with a rate of $1.1\,g\,l^{-1}\,d^{-1}$ under continuous light conditions. Removal of native regulation enables carbon fixation and 2,3-butanediol production in the absence of light. This represents a significant step towards industrial viability and an excellent example of carbon metabolism plasticity.

[1] Department of Chemistry, University of California, Davis, One Shields Avenue, Davis, California 95616, USA. [2] Asahi Kasei Corporation, 2767-11 Niihama, Shionasu, Kojima, Kurashiki, Okayama 711-8510, Japan. Correspondence and requests for materials should be addressed to S.A. (email: satsumi@ucdavis.edu).

Metabolic engineering of photosynthetic organisms allows solar energy to power carbon capture and the production of food, fuels and valuable chemicals[1,2]. Cyanobacteria have attracted much interest as hosts for photosynthetic chemical production due to the simplicity of culture conditions, ease of genetic manipulation and relatively fast cell growth compared to higher plants[3–6]. Several cyanobacterial strains have been engineered for photosynthetic chemical production. However, despite progress in metabolic manipulation and analysis, titres and productivities from cyanobacteria are still far below industrial feasibility[7,8]. Dependency on continuous lighting and the slow process of carbon fixation are particularly limiting[9,10].

We have previously engineered the model cyanobacterium *Synechococcus elongatus* PCC 7942 to produce 2,3-butanediol (23BD) from $CO_2$ and glucose[11–13]. Under natural light, chemical production from an engineered photosynthetic organism would be confined to a limited window of optimal sunlight exposure[13]. However, in order to achieve industrial feasibility, chemical production under both light and dark conditions is essential. As demonstrated in our previous work, concurrent expression of heterologous sugar importers and the 23BD biosynthetic pathway genes allows for chemical production and growth from both $CO_2$ and glucose[13]. The oxidation of sugars provides an increased supply of metabolites and energy independent of photosynthesis, allowing for faster cell growth and continuous chemical production throughout diurnal conditions (12 h light/12 h dark).

Carbon yield is one of the most important factors for economic feasibility and is calculated based on the theoretical maximum yield (TMY). In this study, two substrates, $CO_2$ and glucose, are utilized simultaneously. Owing to their concurrent use and culturing conditions that allow for gas exchange with the environment, it is impossible to directly measure TMY from both substrates. Therefore, we evaluate photomixotrophic production by calculating TMY possible from glucose alone ($0.5\ g_{23BD}/g_{glucose}$). TMY values above 100% indicate 23BD production beyond what is possible from glucose alone, and we assume that these yields reflect the incorporation of $CO_2$ in addition to glucose. It is important to note that glucose utilization is not 100% efficient. The value of the TMY minus the maximum contribution from glucose (100%) represents only the minimum value for $CO_2$ incorporation, and the true contribution is likely higher.

The carbon yield achieved in our previous study was 40% of TMY[12], with great potential for improvement. Thus, the goal of this study is to optimize glucose and $CO_2$ utilization and to improve 23BD production and yield. To accomplish this, we identify and relieve bottlenecks in glucose catabolism, deregulate and enhance $CO_2$ fixation, combine successful modifications and characterize the resulting strain in a variety of production conditions.

Carbon flux in cyanobacteria is often limited by the carbon fixation step in the Calvin Benson (CB) cycle[14]. To overcome this, much work has been done to improve the catalytic activity of the key carbon fixation enzyme, ribulose-1,5-bisphosphate carboxylase/oxygenase (RuBisCO), but with very limited success[15,16]. Several studies have shown that the catalytic efficiency of RuBisCO is already naturally optimized[17,18].

Rather than focusing on improving RuBisCO, we examine carbon metabolism as a whole. Enzyme and metabolite pools of the regenerative phase of the CB cycle are proposed to play a role in determining the overall carbon fixation rate[19,20]. Modifications of other steps in the CB cycle, such as those catalysed by sedoheptulose 1,7-bisphosphatase and transketolase, have been utilized to improve carbon fixation[19–21]. Therefore, we hypothesize that the supplementary carbon source glucose would

increase availability of the substrate for carbon fixation, D-ribulose-1,5-bisphosphate (R15P), and enhance efficiency of the reaction catalysed by RuBisCO.

Here we propose a strategy to increase carbon fixation and chemical production in cyanobacterial grown under light and dark conditions. First, glucose metabolism is rewired through the oxidative pentose phosphate (OPP) pathway to overproduce ribulose-5-phosphate (Ru5P), a precursor of the $CO_2$ fixation pathway. Next, because carbon metabolism in cyanobacteria is precisely controlled in response to environmental changes such as light and carbon availability[22–24], *cp12*, a regulatory gene of the CB cycle, is deleted to amplify conversion of Ru5P to R15P (Fig. 1a,b). This integrated approach of essential gene overexpression and deletion of *cp12* allows for enhanced $CO_2$ fixation, a remarkable increase of 23BD production in both light and dark conditions through light-independent supply of glucose carbons (Fig. 1a,b), and could be applied as a general strategy for improving the efficiency of other photosynthetic organisms.

## Results

**Characterization of glucose catabolism in engineered strains.** To achieve efficient 23BD production from glucose and $CO_2$, we first optimized the genetic construct for the 23BD biosynthesis pathway (Fig. 1c), then paired this optimization with expression of *galP*, which encodes a galactose-proton symporter from *Escherichia coli* (**Strain 3**, Table 1). Details of optimization and a summary of our previous work on 23BD production in *S. elongatus* PCC 7942 can be found in the Supplementary Figs 1, and 2 and Supplementary Note 1. **Strain 3** produced $2.5\ g\,l^{-1}$ of 23BD when glucose was supplied, whereas only $0.4\ g\,l^{-1}$ was produced without glucose (Fig. 2a,b).

To investigate how glucose affects the carbon metabolism of **Strain 3**, we applied metabolomics analysis. Quantification of metabolites from **Strain 3** may explain how the flow of carbon changes in *S. elongatus* because of glucose, which may in turn allow optimization of carbon metabolism and 23BD production of the engineered strains. Cells were grown with or without glucose for 72 h under continuous light. Among the several hundred metabolite signals detected, 136 metabolites were identified and quantified[25,26] (Fig. 1c and Supplementary Data 1) and some key changes to central carbon pathway metabolites were observed. Levels of gluconate and fructose-6-phosphate (F6P) increased by 7.3- and 4.1-fold, respectively, in the presence of glucose (Fig. 1c). In contrast, metabolites of the lower part of the Embden–Meyerhof–Parnas (EMP) pathway, 3-phosphoglycerate (3PGA) and phosphoenolpyruvate (PEP) were lowered by 66% and 88%, respectively, upon the addition of glucose (Fig. 1c). This has also been observed in *Synechocystis sp*. PCC 6803 under similar conditions[27]. Moreover, a decrease of TCA cycle metabolites, including malate, fumarate, succinate and citrate, was observed in glucose-fed cells (Fig. 1c).

In order to understand these data further, we constructed several deletion mutants from **Strain 3** (Table 1) that would separately inactivate one of the branches of carbon metabolism. Three main pathways for glucose catabolism in cyanobacteria were investigated: the OPP pathway, the EMP pathway and the Entner-Doudoroff (ED) pathway (Fig. 1c). Genes encoding the following enzymes were chosen as deletion targets in order to assess the metabolic contribution of each pathway: glucose-6-phosphate dehydrogenase (encoded by *zwf*, **Strain 4**) and 6-phosphogluconate (6PG) dehydrogenase (*gnd*, **Strain 5**) of the OPP pathway, phosphoglucose isomerase (*pgi*, **Strain 6**) and phosphofructokinase (*pfk*, **Strain 7**) of the EMP pathway, and 2-keto-3-deoxygluconate 6-phosphate aldolase (*eda*, **Strain 8**) of the ED pathway (Table 1). Gene replacement with an antibiotic

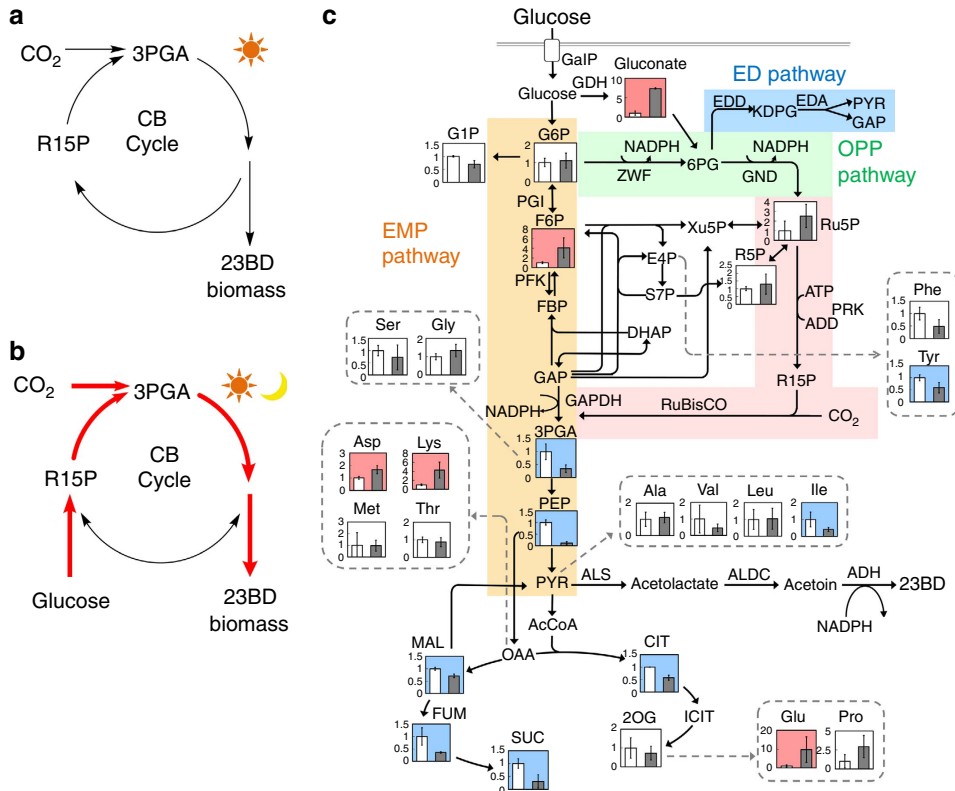

**Figure 1 | Rewiring of carbon metabolism in cyanobacteria.** (**a**) Photoautotrophic conversion of $CO_2$ to 23BD and biomass. (**b**) Coupling glucose metabolism with the CB cycle to enhance $CO_2$ fixation and 23BD production in both light and dark conditions. (**c**) Relative amounts of intracellular metabolites of **Strain 3** (*alsS-alsD-adh + galP*) grown with (grey) and without (white) glucose in continuous light conditions for 72 h where $n = 3$ biological replicates, and error bars represent s.d. Metabolites significantly elevated and decreased with the addition of glucose are labelled red and blue, respectively. ADH, alcohol dehydrogenase; ALDC, acetolactate decarboxylase; ALS, acetolactate synthase; CIT, citrate; DHAP, dihydroxyacetone phosphate; EDA, 2-keto-3-deoxygluconate-6-phosphate aldolase; EDD, 6PG dehydratase; E4P, erythrose-4-phosphate; FBP, fructose-1,6-bisphosphate; FUM, fumarate; F6P, fructose-6-phosphate; GAP, glyceraldehyde-3-phosphate; GalP, galactose-proton symporter; GND, 6PG dehydrogenase; G1P, glucose-1-phosphate; G6P, glucose-6-phosphate; ICIT, isocitrate; KDPG, 2-keto-3-deoxy-6-phosphogluconate; MAL, malate; OAA, oxaloacetate; PEP, phosphoenolpyruvate; PGI, phosphoglucose isomerase; PFK, phosphofructokinase; PRK, phophoribulokinase; PYR, pyruvate; AcCoA, acetyl-CoA; RuBisCO, ribulose-1,5-bisphosphate carboxylase/oxygenase; R5P, ribose-5-phosphate; R15P, ribulose-1,5-bisphosphate; Ru5P, ribulose-5-phosphate; SUC, succinate; S7P, sedoheptulose-7-phosphate; Xu5P, xylulose-5-phosphate; ZWF, G6P dehydrogenase; 2OG, 2-oxoglutarate; 23BD, 2,3-butanediol; 3PGA, 3-phosphoglycerate; and 6PG, 6-phosphogluconate.

resistance gene and complete segregation of **Strains 4** ($\Delta zwf$) and **7** ($\Delta pfk$) were verified by PCR and sequencing. However, despite repeated trials, complete deletion of *gnd*, *pgi* and *eda* from the genome could not be achieved, indicating that these genes were essential in our culture conditions. However, we chose to include these strains in further tests to characterize the effects of partial knockouts of each gene. It is of note, however, that *gnd* has been successfully deleted in wild-type *S. elongatus*[28,29]. We expected that disruption of pathways responsible for glucose metabolism would result in a defect in growth and/or 23BD production. The photomixotrophic growth of **Strain 4** ($\Delta zwf$) was impaired after 72 h (Fig. 2a). Also, 23BD production of **Strain 4** decreased by 62% compared to **Strain 3** when grown with glucose (Fig. 2b). However, other mutants showed no significant changes in growth or 23BD production compared to **Strain 3** (Fig. 2a,b). Impairment of growth and 23BD production of **Strain 4** suggested that glucose is metabolized through the OPP pathway, but if that were the case, then we would expect **Strain 5** to also display defects. That **Strain 5** (or **Strains 6–8**) did not show growth or 23BD production defects makes it uncertain which of the three pathways was primarily responsible for metabolizing fed glucose. Also, it is possible that pathway disruption was compensated for by upregulation of other

glucose metabolism pathways. Therefore, we decided to individually overexpress genes of all three pathways in order to test their influence on 23BD production.

**Guiding glucose flux in central carbon metabolism.** To determine which pathway is responsible for glucose metabolism, and to explore the benefits of directed glucose flux, we overexpressed the key genes of each pathway (**Strains 9**, *galP-zwf-edd* (ED); **10**, *galP-pgi* (EMP); **11**, *galP-zwf-gnd* (OPP), Table 1 and Supplementary Fig. 2). *In vitro* enzyme assays confirmed the functional expression of each genes (Supplementary Table 4). Enzymatic activities of ZWF and GND were 14- and 3.6-fold higher, respectively, in **Strain 11** than in **Strain 3** (without overexpression) (Supplementary Table 4). **Strain 10** showed a large increase in activity of PGI, whereas no significant activity was detected in **Strain 3** (Supplementary Table 4). Therefore, *in vivo* experiments were conducted to test the effects of these overexpressed genes on growth and 23BD production. Strains were grown with the addition of glucose under continuous light for 72 h (Fig. 2c–e). We expected an increase in growth, 23BD production and/or increased glucose consumption to be correlated with overexpression of an active glucose metabolism

**Table 1 | Strains used in this study.**

| Strains | Strain No. | Genotype | References |
|---|---|---|---|
| AL257 | | *Synechococcus elongatus* PCC 7942 | Golden[43] |
| AL2491 | 1 | AL257 + NSIII:: *lacI$^q$*; Ptrc: *alsD-alsS-adh*; *gent$^R$* | This study |
| AL2253 | 2 | AL257 + NSIII:: *lacI$^q$*; P$_L$lacO$_1$: *alsD-alsS-adh*; *gent$^R$* | 47 |
| AL2456 | 3 | **1** + NSI:: *lacI$^q$*; Ptrc: *galP*; *spec$^R$* | This study |
| AL2799 | 4 | **3** + *zwf*:: *kan$^R$* | This study |
| AL2800 | 5 | **3** + *gnd/Δgnd*:: *kan$^R$* | This study |
| AL2798 | 6 | **3** + *pgi/Δpgi*:: *kan$^R$* | This study |
| AL2680 | 7 | **3** + *pfk*:: *cm$^R$* | This study |
| AL2801 | 8 | **3** + *eda/Δeda*:: *kan$^R$* | This study |
| AL2894 | 9 | **1** + NSI:: *lacI$^q$*; Ptrc: *galP-zwf-edd*; *spec$^R$* | This study |
| AL2937 | 9-2 | **9** + NSII:: *lacI$^q$*; Ptrc: *eda*; *kan$^R$* | This study |
| AL2895 | 10 | **1** + NSI:: *lacI$^q$*; Ptrc: *galP-pgi*; *spec$^R$* | This study |
| AL2936 | 10-2 | **10** + NSII:: *lacI$^q$*; Ptrc: *pfkA*; *kan$^R$* | This study |
| AL2896 | 11 | **1** + NSI:: *lacI$^q$*; Ptrc: *galP-zwf-gnd*; *spec$^R$* | This study |
| AL2556 | 12 | **3** + *cp12*:: *cm$^R$* | This study |
| AL2568 | 13 | **3** + *cp12*:: *lacI$^q$*; Ptrc: *rbcLXS*; *kan$^R$* | This study |
| AL2847 | 14 | **3** + *cp12*:: *lacI$^q$*; Ptrc: *prk-rbcLXS*; *kan$^R$* | This study |
| AL2935 | 15 | **11** + *cp12*:: *lacI$^q$*; Ptrc: *prk-rbcLXS*; *kan$^R$* | This study |
| AL1793 | 16 | AL257 + NSIII:: *lacI$^q$*; P$_L$lacO$_1$: *sfgfp*; *gent$^R$* | 47 |
| AL2575 | 17 | AL257 + NSIII:: *lacI$^q$*; Ptrc: *sfgfp*; *gent$^R$* | This study |

pathway. **Strain 9** (*galP-zwf-edd*) showed similar growth and glucose consumption rates compared to **Strain 3** (*galP*), but decreased 23BD production (Fig. 2c–e). By contrast, the growth rates of **Strains 10** (*galP-pgi*) and **11** (*galP-zwf-gnd*) were enhanced by 87% and 82%, respectively (Fig. 2c), and the glucose consumption rates of these strains increased by 118% and 129%, respectively, compared to **Strain 3** (Fig. 2e). This provides strong evidence that glucose is metabolized through both the EMP and the OPP pathways. However, the 23BD production of **Strains 10** and **11** was reduced compared to that of **Strain 3** (Fig. 2d), suggesting that a larger portion of carbon flux from glucose was utilized for biomass formation rather than 23BD production in **Strains 10** and **11**. To improve glucose consumption by the ED pathway, we modified **Strain 9** by additionally expressing the downstream gene, *eda*, resulting in **Strain 9–2**. Similarly, we expressed *pfkA* from the EMP pathway, resulting in **Strain 10-2** (Table 1). However, neither glucose consumption nor 23BD production was improved in **Strains 9–2** and **10–2** compared to the parent strains (Supplementary Fig. 3). Because glucose metabolism was improved in **Strains 10** and **11** (Fig. 2e), we next focused on redirecting carbon flux towards 23BD production rather than biomass formation.

**Redirection of carbon flux towards 23BD biosynthesis**. We also wanted to explore whether modification to the CB cycle and its regulation would be beneficial for our production strain.

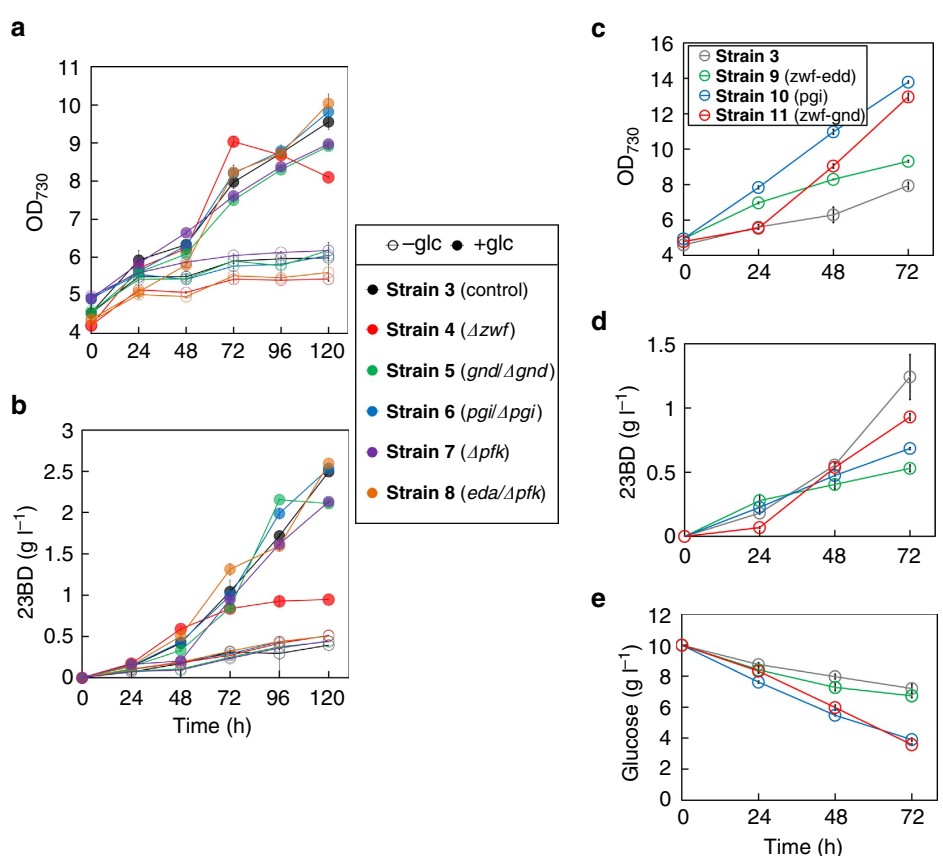

**Figure 2 | Characterization and activation of glucose metabolism.** Cells were cultured in 10 ml of BG11 media containing 10 g l$^{-1}$ glucose and 20 mM NaHCO$_3$ in continuous light conditions. IPTG (0.1 mM) was added at 0 h. Cell growth (**a**) and 23BD concentration (**b**) profiles of **Strains 3** (black), **4** (*Δzwf*, red), **5** (*gnd/Δgnd*, green), **6** (*pgi/Δpgi*, blue), **7** (*Δpfk*, purple) and **8** (*eda/Δeda*, orange). (**c**–**e**) Cell growth (**c**), 23BD concentration (**d**) and glucose concentration (**e**) profiles of **Strains 3** (grey), **9** (*galP-zwf-edd*, green), **10** (*galP-pgi*, blue) and **11** (*galP-zwf-gnd*, red). *N* = 3 biological replicates; error bars represent s.d.

Specifically, we sought to direct more carbon flux through the CB cycle by overexpressing crucial genes of the pathway and removing native regulation of the cycle. CP12 is a small regulatory protein that represses two important enzymes of the CB cycle, glyceraldehyde-3-phosphate dehydrogenase (GAPDH) and phosphoribulokinase (PRK). We hypothesized that the deletion of *cp12* could be effective for redirecting the carbon flux through the CB cycle towards 23BD production. CP12 is activated in response to an elevated NAD(H)/NADP(H) ratio, a condition that was previously observed in photomixotrophic[27] and dark[22] conditions. Once activated, CP12 inhibits GAPDH, which controls the carbon flux towards the lower EMP pathway in cyanobacteria[30]. CP12 also inhibits PRK, which is responsible for the conversion of Ru5P to R15P, the substrate for the $CO_2$ fixation by RuBisCO (Fig. 1c). Thus, the deletion of *cp12* should increase carbon flux to the CB cycle, and may increase the rate of $CO_2$ fixation, both of which should increase the intracellular pool of pyruvate and lead to an improvement in 23BD production.

To test whether changing regulation of carbon metabolism could increase carbon flow to 23BD biosynthesis, we constructed three strains. We disrupted the *cp12* gene in **Strain 3** to understand whether the absence of *cp12* alone is sufficient for metabolic flux redirection. We also disrupted *cp12* and inserted one of two sets of essential genes involved in carbon fixation: *rbcL-rbcX-rbcS* (*rbcLXS*), which encodes the RuBisCO subunits and their chaperone from *Synechococcus* sp. PCC 7002, and *rbcLXS* plus *prk*, which encodes PRK from *S. elongatus* (Supplementary Fig. 2). Despite multiple attempts, a strain with $\Delta cp12$:: *prk* and no additional *rbcLXS* could not be successfully constructed. *Synechococcus* sp. PCC 7002 is known as one of the fastest-growing cyanobacteria[31]; therefore, we hypothesized that this particular RuBisCO would show desirable enzyme kinetics. We expected that overexpression of *rbcLXS* and *prk* independent of native regulation would increase carbon flow through these bottleneck steps of the CB cycle, providing additional carbon for 23BD biosynthesis. These modifications resulted in **Strains 12** ($3 + \Delta cp12$), **13** ($3 + \Delta cp12$:: *rbcLXS*) and **14** ($3 + \Delta cp12$:: *prk-rbcLXS*; Table 1). Amplified activity of PRK (1.3-fold) was confirmed in **Strain 14** compared to **Strain 3**; however, there was no significant difference in RuBisCO activity between **Strain 13** or **14** and **Strain 3** (Supplementary Table 4).

We hypothesized that these strains would result in improvements in 23BD production by increasing $CO_2$ incorporation. Thus, $^{13}C$-labelled substrates were used to observe differences in the ratio of 23BD derived from $CO_2$ versus fed glucose. **Strains 3, 12, 13** and **14** were monitored for 23BD production after growth with $10\,g\,l^{-1}$ of U-$^{13}C$ glucose and 20 mM of unlabelled NaHCO$_3$ under continuous light. Assuming that all carbons of 23BD originated from either glucose or $CO_2$, we measured the percentage of carbons in 23BD derived from either glucose or $CO_2$ (Fig. 3a). In **Strain 3,** 35% of 23BD carbons were derived from $CO_2$, while all three of the altered regulation strains showed an increase in 23BD carbons derived from $CO_2$: **Strain 12** ($\Delta cp12$) had 53% of 23BD carbons derived from $CO_2$, **Strain 13** ($\Delta cp12$:: *rbcLXS*) had 39% and **Strain 14** ($\Delta cp12$:: *prk-rbcLXS*) had 53% (Fig. 3a). Interestingly, although these data suggest that carbon flow was successfully redirected to carbon fixation, these modifications did not improve cell growth or 23BD biosynthesis (Fig. 3b,c). Also, since there was no significant difference in RuBisCO activity between **Strains 3** and **12** (Supplementary Table 4), the beneficial phenotype is likely due to the enhanced enzymatic activity of PRK in **Strain 14** and not that of RuBisCO.

In the absence of light, CP12 blocks the PRK-catalysed conversion of Ru5P into R15P (ref. 22), preventing $CO_2$ fixation. Removing this regulation by deletion of *cp12* should allow for $CO_2$ fixation to occur regardless of light conditions.

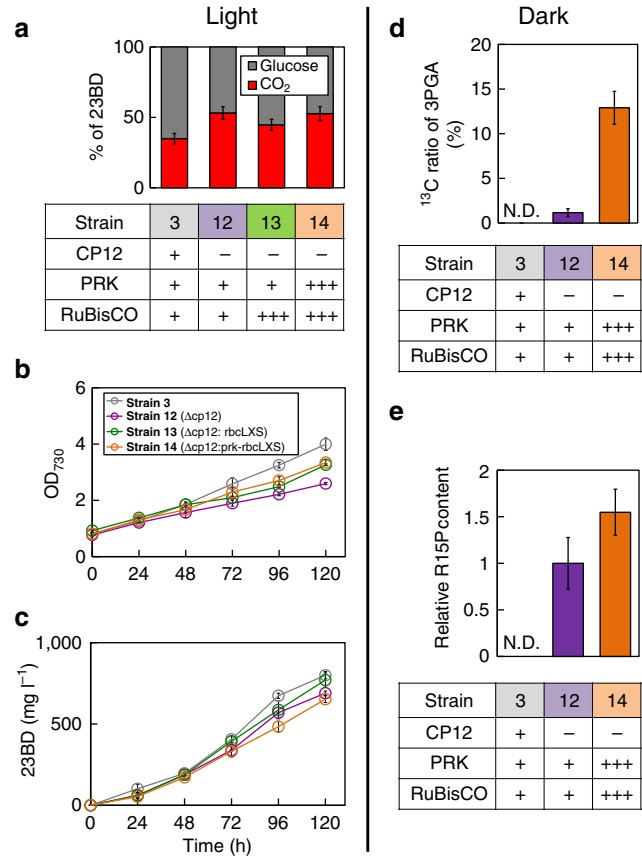

**Figure 3 | Redirection of carbon flux towards $CO_2$ fixation in light and dark conditions.** (**a–c**) Cells were cultured in 10 ml of BG11 media containing $10\,g\,l^{-1}$ U-$^{13}C$ glucose and 20 mM unlabelled NaHCO$_3$ in continuous light conditions. '**-**', '**+**' and '**+ + +**' indicate that each gene in the corresponding row is deleted, natively expressed or overexpressed, respectively. (**a**) The percentage of 23BD produced from either glucose or $CO_2$ by each strain. Growth (**b**) and 23BD concentration (**c**) profiles of **Strains 3** (grey), **12** ($\Delta cp12$, purple), **13** ($\Delta cp12$:: *rbcLXS*, green) and **14** ($\Delta cp12$:: *prk-rbcLXS*, orange). (**d,e**) **Strains 3**, **12** and **14** were cultured in 10 ml of BG11 media containing $10\,g\,l^{-1}$ unlabelled glucose and $^{13}C$-NaHCO$_3$ for 24 h under continuous dark conditions. $^{13}C$ labelling ratio of intracellular 3PGA (**d**) and concentration of intracellular R15P normalized with that of **Strain 12** (**e**). $N = 3$ biological replicates; error bars represent s.d. N.D. indicates not detectable.

To test our hypothesis that these strains could grow and produce 23BD independent of light, **Strains 3, 12** ($\Delta cp12$) and **14** ($\Delta cp12$:: *prk-rbcLXS*) were grown with $10\,g\,l^{-1}$ glucose and 20 mM $^{13}C$-NaHCO$_3$ in complete darkness for 24 h, and the $^{13}C$ ratio of intracellular 3PGA was determined. (**Strain 13** was not tested since it showed very little improvement of carbon fixation compared to **Strains 12** and **14**.) 3PGA is a direct product of the reaction catalysed by RuBisCO, and so labelled 3PGA would be clear evidence of carbon fixation. In the 3PGA from **Strain 14**, 12.6% of the carbons were labelled with $^{13}C$. Very few carbons were labelled in **Strain 12** and no labelled carbons were detected in **Strain 3** (Fig. 3d). These results indicate that deletion of *cp12* permitted limited $CO_2$ fixation in darkness, and deletion of *cp12* combined with overexpression of *prk* (and *rbcLXS*) resulted in substantial $CO_2$ fixation by RuBisCO in darkness. As further evidence of enhanced carbon flux to the $CO_2$ fixation pathway, we also observed a 1.5-fold increase in the intracellular concentration of the RuBisCO substrate, R15P, in **Strain 14** compared to **Strain 12**, and no accumulation of R15P was observed in **Strain 3**

(Fig. 3e). That these strains show altered concentrations of this key carbon fixation metabolite supports our conclusion that $CO_2$ fixation was enabled in **Strains 12** and **14** in the absence of light.

**Coupling glucose metabolism and $CO_2$ fixation.** With the success of our engineering strategies for increased glucose consumption (by overexpression of the EMP or the OPP pathway, **Strains 10** (*galP-pgi*) and **11** (*galP-zwf-gnd*)), and increased carbon fixation (by deletion of *cp12* and overexpression of *prk* and *rbcLXS*, **Strain 14** ($\Delta cp12$:: *prk-rbcLXS*)), we hypothesized that a combination of these would produce a synergistic effect to further improve 23BD production and $CO_2$ fixation. Since the OPP pathway is the most direct route from glucose to $CO_2$ fixation, we chose to focus on this pathway for further optimization (**Strain 11** (*galP-zwf-gnd*)). In two steps, it converts glucose-6-phosphate (G6P) to Ru5P, which can then be directed into the $CO_2$ fixation pathway through the reaction catalysed by PRK (Fig. 1c). Thus, in **Strain 11**, we replaced *cp12* with *prk* and *rbcLXS*, (**Strain 15**, Table 1). **Strains 3** (*galP*), **11** (*galP-zwf-gnd*), **14** ($3 + \Delta cp12$:: *prk-rbcLXS*) and **15** ($11 + \Delta cp12$:: *prk-rbcLXS*) were cultured with $10 \text{ g l}^{-1}$ of glucose and 20 mM NaHCO$_3$ for 48 h, and growth, glucose consumption and 23BD production were quantified (Fig. 4). The growth and glucose consumption rates of **Strains 11** and **15** were roughly three times higher than those of **Strains 3** and **14** (Fig. 4a,c), and a remarkable improvement of 23BD production ($1.4 \text{ g l}^{-1}$ at 48 h compared to $0.5 \text{ g l}^{-1}$ in **Strain 3**) was observed in **Strain 15** (Fig. 4b).

To analyse the mechanism of the drastic improvement of 23BD production observed in **Strain 15** (*galP-zwf-gnd* + $\Delta cp12$::

*prk-rbcLXS*), metabolomics analysis was applied to **Strains 3, 11** and **15** (Fig. 4d, Supplementary Data 2 and Supplementary Fig. 4). Gluconate, 3PGA and PEP were markedly reduced in **Strains 11** and **15** compared to **Strain 3**, whereas a reduction of G6P, F6P, Ru5P and ribose-5-phosphate was only observed in **Strain 15** (Fig. 4d). These data suggest that carbon flow through these steps of glucose metabolism towards the CB cycle has been streamlined by overexpression of the OPP pathway in **Strain 15**. In particular, the large reduction ($\sim 90\%$) of Ru5P in **Strain 15** compared to **Strains 3** and **11** suggests that carbons were successfully redirected to the carbon fixation pathway by the action of PRK. Also, the significant increase of R15P, the substrate of $CO_2$ fixation, in **Strain 15** (Fig. 4e) suggests that RuBisCO is the bottleneck in an otherwise streamlined metabolism of glucose.

It would be useful to quantify carbon flux as a way to directly compare the carbon fixation efficiency of these engineered strains. 23BD is easily quantified; however, because of the decarboxylation steps in the 23BD biosynthesis pathway, 23BD is not an accurate measure of newly fixed carbon (Supplementary Fig. 5). Therefore, the intracellular concentration of R15P was determined to evaluate the relative carbon flux in the carbon fixation pathway. It is very likely that the concentrations of Ru5P and R15P were lower than the $K_m$ values of PRK (270 μM; ref. 32) and RuBisCO (27 μM; ref. 33), respectively. Thus, the increase of R15P indicates an enhanced carbon flux in the pathway. R15P content of **Strain 15** (*galP-zwf-gnd* + $\Delta cp12$:: *prk-rbcLXS*) was increased by 4.6-fold compared to **Strain 3** (*galP*; Fig. 4e). By contrast, there was no significant difference between **Strains 3** and **11** (*galP-zwf-gnd*). In addition, specific production of

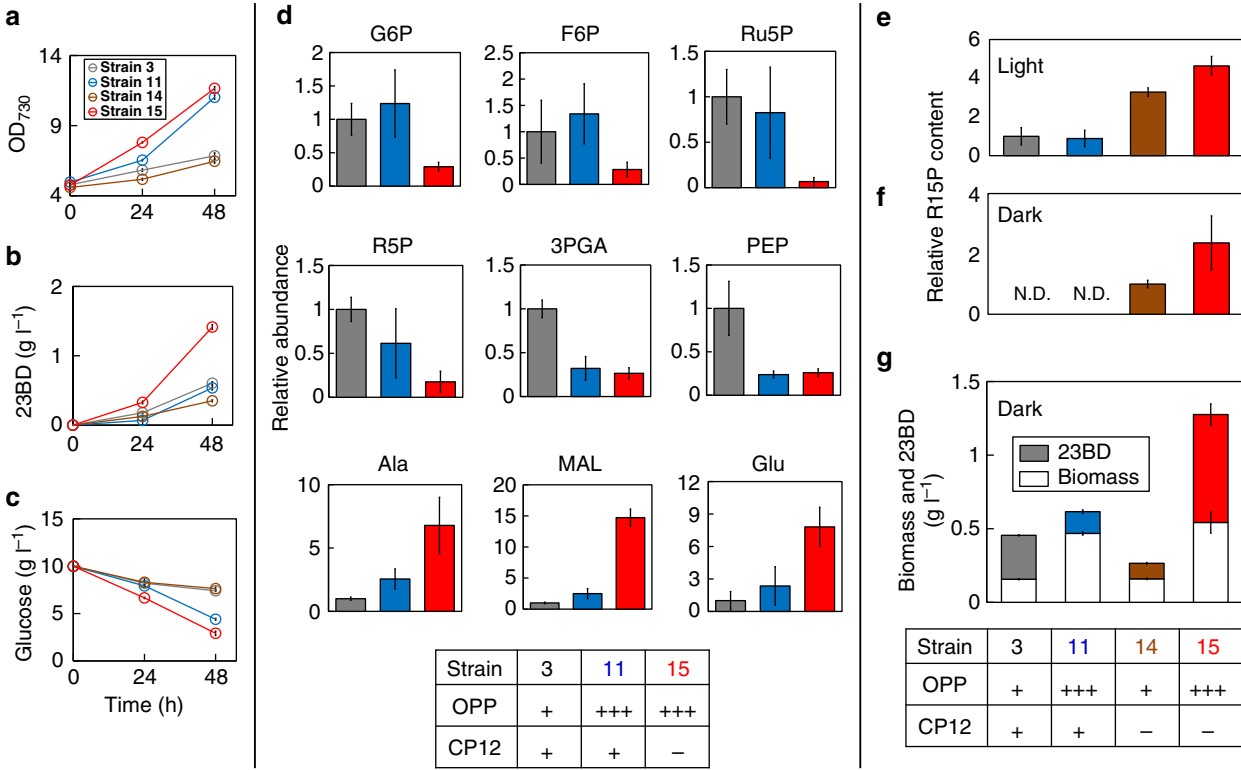

**Figure 4 | Coupling the OPP pathway and the $CO_2$ fixation pathway.** Cells were cultured in 10 ml of BG11 media containing $10 \text{ g l}^{-1}$ glucose and 20 mM NaHCO$_3$ in continuous light conditions (**a–e**) and dark conditions (**f,g**). OPP = with ($+++$) and without ($+$) overexpression of *zwf* and *gnd*; CP12 = *cp12* gene was intact ($+$) and deleted ($-$) with *prk* and *rbcLXS*. Metabolite abbreviations are the same as those used in Fig. 1. Growth (**a**), 23BD concentration (**b**) and glucose concentration (**c**) profiles of **Strains 3** (grey), **11** (*galP-zwf-gnd*, blue), **14** ($\Delta cp12$:: *prk-rbcLXS*, brown) and **15** ($11 + \Delta cp12$:: *prk-rbcLXS*, red). Intracellular concentrations of metabolites (**d**) of **Strain 3, 11** and **15** at 48 h. Relative content of intracellular R15P of **Strains 3, 11, 14** and **15** grown in continuous light (**e**) and dark (**f**) conditions for 24 h. Cell biomass and 23BD production (**g**) of the same strains grown in continuous dark conditions for 24 h. $N = 3$ biological replicates; error bars represent s.d.' N.D. indicates not detectable.

intracellular alanine from $CO_2$ was examined by feeding $^{13}C$-$NaHCO_3$ and unlabelled glucose. Alanine is generated from pyruvate by a single reaction catalysed by alanine transaminase. Thus, alanine's intracellular pool size can be correlated to carbon flux towards pyruvate and 23BD productivity. An increase of twofold in $^{13}C$-labelled alanine was observed in **Strain 15** compared to **Strains 3** and **11** (Supplementary Fig. 6), supporting the conclusion that carbon flux through the CB pathway was enhanced.

Notable changes in metabolites related to biomass formation were also observed that may at least partially explain the increased 23BD production of **Strain 15**. A reduction of metabolites required for purine and pyrimidine biosynthesis, such as adenine and thymidine 5-phosphate (dTTP), and those required for lipid production, such as glycerol-3-phosphate, was observed in **Strain 15** (Supplementary Fig. 4) but not in **Strain 11**. Since these metabolites are associated with biomass formation, their decrease may reflect a change in carbon partitioning between **Strains 11** and **15**. Another possibility is that these metabolites have a high in/out flux as a result of rapid growth, and are at a lower steady-state concentration. This has been observed in flux analysis of fast-growing cultures[34–36]. However, the lower concentrations of these metabolites in **Strain 15** compared to **Strain 11** did not correlate with a faster growth rate, only an increase in 23BD production (Fig. 4a,b). On the other hand, levels of the TCA cycle intermediates such as malate, fumarate and amino acids derived from either pyruvate or 2-oxoglutarate were elevated in **Strain 15** compared to **Strain 3** (Fig. 4d and Supplementary Fig. 4). Malate is known as the main substrate for pyruvate biosynthesis in cyanobacteria[37]. Since pyruvate is a key precursor to 23BD biosynthesis, a 15-fold increase in malate is consistent with the remarkably enhanced 23BD production observed in **Strain 15**.

**Effects of metabolic rewiring on 23BD production in darkness.** When glucose is metabolized via the OPP pathway, NADPH production is balanced by oxidization steps in 23BD biosynthesis, regardless of whether carbon is metabolized by $CO_2$ fixation or the lower EMP pathway (Fig. 1c). This makes coupling the OPP

and $CO_2$ fixation pathways desirable in terms of increasing NADPH availability for growth and chemical production, especially under periods of continuous darkness when NADPH is starkly limiting. Therefore, **Strain 15** (*galP-zwf-gnd* + $\Delta$*cp12*:: *prk-rbcLXS*) should be able to generate sufficient reducing power (NADPH) to maintain 23BD production in the absence of light. To test this hypothesis, strains were tested for 23BD production in continuous dark conditions.

**Strains 3** (*galP*), **11** (*galP-zwf-gnd*), **14** ($\Delta$*cp12*:: *prk-rbcLXS*) and **15** (**11** + $\Delta$*cp12*:: *prk-rbcLXS*) were cultured with 20 mM $NaHCO_3$ and 10 g l$^{-1}$ glucose for 24 h in complete darkness. We expected that the combined overexpression of the OPP pathway and deregulation of GAPDH and PRK in **Strain 15** would lead to more efficient 23BD production in dark conditions. R15P was elevated twofold in **Strain 15** compared to **Strain 14**, while R15P was not detected in **Strains 3** and **11** (Fig. 4f), indicating that deletion of *cp12* allowed an increase in CB cycle carbon pools even in the dark. The biomass and 23BD production of **Strain 15** were enhanced by 2.9- and 2.5-fold, respectively, compared to **Strain 3**, while **Strain 11** only showed a 2.7-fold increase in biomass production (Fig. 4g). Also, although **Strain 14** was able to fix carbon in darkness, as evidenced by the production of $^{13}C$-labelled 3PGA, its biomass and 23BD produced were lower than those of **Strain 3** (Figs 3d and 4g). These data show that the *cp12* deletion was effective in activating the $CO_2$ fixation pathway by deregulating PRK, and that the coupling with OPP overexpression redirected carbon flux to synergistically increase biomass and 23BD production.

To determine whether carbon fixation was also enhanced in **Strains 3**, **11**, **14** or **15**, cells were cultured with $^{13}C$-$NaHCO_3$ and unlabelled glucose for 24 h in continuous darkness, and $^{13}C$-labelled 23BD was quantified. A characteristic mass signal from $^{13}C$-$NaHCO_3$ was detected at 180 *m/z*, originating from either M + 2, M + 3 or M + 4 of derivatized 23BD produced by **Strain 15**, and this signal was not observed from **Strains 3**, **11** or **14** (Supplementary Fig. 7). This mass spectrum of 23BD produced by **Strain 15** indicated that 14% of 23BD carbons came from $CO_2$. This result provides direct evidence that the CB cycle was active in **Strain 15** even in darkness.

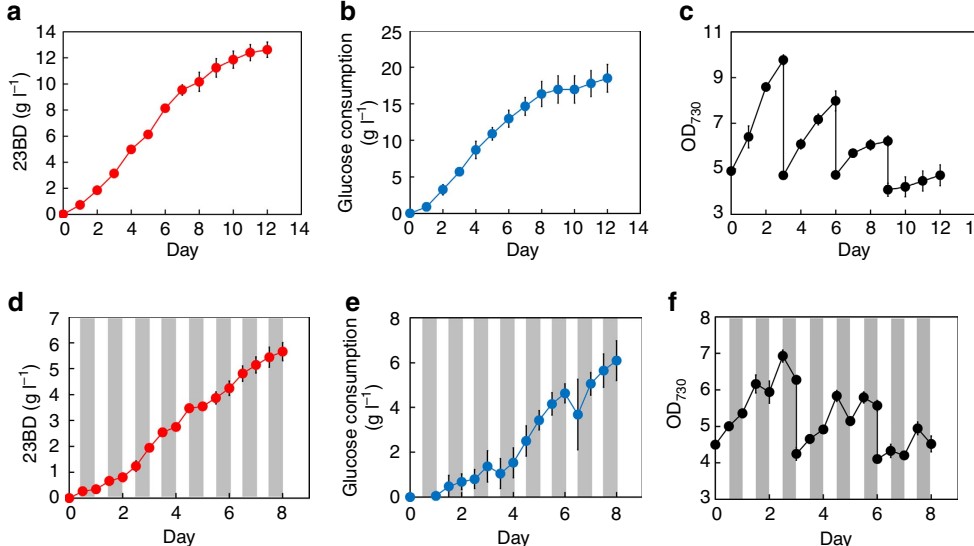

**Figure 5 | Long-term production in continuous light and diurnal light conditions.** Strain 15 (**1** + *galP-zwf-gnd* + $\Delta$*cp12*:: *prk-rbcLXS*) was cultured in 10 ml BG11 media containing 15 g l$^{-1}$ glucose and 20 mM $NaHCO_3$ in continuous light conditions (**a–c**) and diurnal light conditions (**d–f**). Grey shades represent dark phases. Cells were induced with 1 mM IPTG on day 0. On days 3, 6 and 9, cells were collected by centrifugation and resuspended at an $OD_{730}$ of 5.0 in fresh production media. 23BD production (**a,d**), glucose consumption (**b,e**) and growth (**c,f**) profiles of **Strain 15** where $N = 3$ biological replicates, and error bars represent s.d.

**Long-term production of 23BD in continuous light conditions**. For industrialization of photosynthetic chemical production, maximization of production titre is crucial since it greatly affects production recovery costs. To demonstrate that our engineering strategies yielded a strain able to produce high concentrations of 23BD over the long term, we cultured **Strain 15** (*galP-zwf-gnd* + Δ*cp12*:: *prk-rbcLXS*), the leading strain, with $15\,g\,l^{-1}$ glucose and 20 mM NaHCO$_3$ for 12 days under continuous illumination. To maximize production, we modified culture conditions (Supplementary Fig. 8), hypothesizing that increased isopropyl-β-D-thiogalactoside (IPTG) and A5 trace metal ($Mn^{2+}$, $Co^{2+}$, $Cu^{2+}$ and $MoO_4^{-2}$) concentrations would improve gene expression and cofactor availability. Cumulative 23BD production of **Strain 15** was $12.6\,g\,l^{-1}$ on day 12, while total consumption of glucose was $18.5\,g\,l^{-1}$ (Fig. 5a–c). The yield was 136% of the TMY of 23BD calculated from glucose alone ($0.5\ g_{23BD}/g_{Glucose}$). That the yield is above 100% indicates that both $CO_2$ and glucose were successfully utilized with at least 36% of carbons derived from $CO_2$. Given that the yield of 23BD in photomixotrophic conditions reported in our previous study was only 40% of the TMY[13], the efficiency of production achieved in this study represents a dramatic improvement.

**Long-term production of 23BD in diurnal light conditions**. To show that **Strain 15** is also able to produce 23BD for an extended period of time without continuous illumination, we next cultured **Strain 15** under diurnal light (12 h light/12 h dark) conditions with $15\,g\,l^{-1}$ glucose and 20 mM NaHCO$_3$ (Fig. 5d–f). Although optical density decreased during many of the dark phases (Fig. 5f), 23BD production lasted for 8 days, and was observed regardless of light availability, reaching a final titre of $5.7\,g\,l^{-1}$ (Fig. 5d). This production, and consumption of $6.1\,g\,l^{-1}$ glucose (Fig. 5e), corresponds to 195% of the TMY.

### Discussion

Here we describe a strategy to improve $CO_2$ fixation and chemical production by rewiring carbon metabolism in cyanobacteria. We increased carbon flux from glucose by enhancing the OPP pathway, and then redirected it towards the carbon fixation step catalysed by RuBisCO. The engineered strain, **Strain 15**, showed a variety of advantages, including remarkable improvement of 23BD production and enhanced $CO_2$ fixation in both light and dark conditions.

Our integrative approach also explored improving phototrophic carbon fixation. We used glucose supplementation to boost R15P supply (Fig. 1b), and deleted *cp12* to increase PRK-catalysed conversion of Ru5P into R15P. As expected, R15P content was the highest in **Strain 15** grown with glucose in both light and dark conditions (Fig. 4e,f), suggesting that carbon flux in the $CO_2$ fixation pathway was significantly improved. These results also demonstrate the synergistic effects of deregulating carbon fixation and enhancing glucose catabolism for biomass and chemical production.

Improved carbon fixation in **Strain 15** was also beneficial for long-term production under continuous and diurnal light conditions. With continuous lighting, we demonstrated a 23BD production titre of $12.6\,g\,l^{-1}$ from $18.5\,g\,l^{-1}$ glucose in 12 days. This yield (136%) greatly exceeded the TMY, indicating simultaneous use of glucose and $CO_2$. Unexpectedly, yield from diurnal light conditions was even more efficient: $5.7\,g\,l^{-1}$ 23BD produced from $6.1\,g\,l^{-1}$ glucose in 8 days, which is 195% of TMY. This suggests that $CO_2$ fixation and recycling were key to improving carbon yield, even during dark phases, and can be partially explained by changes in the carbon flux through the CB cycle during darkness (Supplementary Fig. 7). Furthermore, cell

growth in diurnal conditions was decreased compared to constant light conditions (Fig. 5), suggesting that yield was improved by partitioning carbon to 23BD production rather than biomass formation.

An alternative strategy to photomixotrophic production has been previously demonstrated: a $CO_2$ fixation pathway consisting of PRK and RuBisCO was introduced into the heterotrophic organism, *Saccharomyces cerevisiae*, to improve ethanol yield[38]. However, yield from galactose was only increased by 8% ($0.44\ g_{Ethanol}/g_{Galactose}$), and is still below the TMY ($0.51\ g_{Ethanol}/g_{Galactose}$). Succinate production from glycerol via carboxylation has been demonstrated to 133% of the TMY[39,40] and the experimental carbon yield was 120% (ref. 41). However, carboxylation exclusive to a particular biosynthetic pathway is limited in applicability. By contrast, RuBisCO carboxylation is connected to central carbon metabolism, and thus to a range of chemical production pathways. Indeed, our method for improvement of carboxylation could be applied to increase photosynthetic production of a wide range of chemicals.

The strategy utilized in our study improved the productivity, titre and yield of 23BD production (Fig. 5). Although cyanobacteria have been engineered to produce a number of compounds, the productivities and titres are much too low (most on the order of $mg\,l^{-1}$) to commercialize these technologies (Supplementary Fig. 9). In addition, strains developed for photoautotrophic production are restricted by light availability, even though dense cultures ($OD_{730} > 5$) are often required for efficient production. This presents a challenge since light penetrance is hindered by mutual cell shading, resulting in decreased light availability per cell, and arrested cell growth and chemical production. In addition, in natural lighting conditions the active growth and production period falls to 8–14 h per day. Our approach overcomes these obstacles, providing significant improvements to chemical production in a variety of industrially relevant lighting and growth conditions. Economic analysis of 23BD production in diurnal light conditions using a semi-open pond illuminated by natural sunlight showed that operation profit could be increased fivefold in photomixotrophic conditions compared to photoautotrophic conditions (Supplementary Fig. 10 and Supplementary Note 2). This is despite the additional production costs required for photomixotrophic cultures, such as feedstock cost and steam sterilization of production media. In light conditions, it is advantageous to maximize cell density for chemical production. Sugar supplementation in these conditions may prove useful for overcoming density-dependent light deficiencies in phototrophic cultures. The described approach allows the engineered strains to produce and grow without light, capitalizing on night time hours. Glucose supplementation would also avoid issues with sunlight variability. In addition, the engineered strain's high titre and productivity provide groundwork for production in smaller surface area bioreactors relative to outdoor photoautotrophic production. The recent development of less expensive light-emitting diode (LED) can provide optimal light intensity and wavelength for an energy-efficient production system. (Stored electricity generated from a renewable source, such as a solar panel or wind turbine, would be utilized to power the light-emitting diode.) Faster growth in a smaller surface area system may reduce the risk of contamination compared to current outdoor systems. Hence, this study demonstrates the many advantages of a next-generation scheme of cyanobacterial chemical production with potential for broad applicability in industrial production systems.

### Methods

**Reagents**. The following reagents were obtained from Sigma-Aldrich: glucose, 1,3-propanediol, phenylboronic acid, cycloheximide, 23BD, acetoin, pyruvate, PEP,

6PG, G6P, F6P, Ru5P, R15P, ATP, dithiothreitol (DTT), NADH, NADPH, NADP$^+$, phosphocreatine, RuBisCO, pyruvate kinase, lactate dehydrogenase, G6P dehydrogenase, 3PGA kinase, GAPDH and creatine kinase. U-$^{13}$C glucose and $^{13}$C-NaHCO$_3$ were obtained from Cambridge Isotope Laboratories. IPTG and chloramphenicol were obtained from Fischer Scientific. Gentamycin was purchased from Teknova. Spectinomycin was purchased from MP Biomedicals. Kanamycin was purchased from IBI Scientific. Phusion polymerase was purchased from New England Biolabs. All oligonucleotide synthesis and DNA sequencing were performed by Eurofins MWG Operon Inc.

**Plasmid construction.** All primers and plasmids used in this study are listed in Supplementary Tables 1 and 2, respectively. The target genes and vector fragments used to construct plasmids were amplified using PCR with the primers and templates described in Supplementary Table 3. The resulting fragments were assembled by sequence and ligation-independent cloning[42].

**Strain construction.** Strains used in this study are listed in Table 1. For transformation of *S. elongatus*[43], cells at OD$_{730}$ ~0.4 was collected from 2 ml of culture by centrifugation, washed and concentrated in 100 μl of fresh BG11 medium. Plasmid DNA (5 μg) was added to the cells. The tube was incubated overnight at 30 °C. Cells were plated on a BG11 plate containing appropriate antibiotics and incubated at 30 °C under constant light until colonies appear. A schematic of gene integration is summarized in Supplementary Fig. 2. Complete chromosomal segregation for the introduced fragments was achieved through propagation of multiple generations on selective agar plates. Correct recombinants were confirmed by colony PCR and sequencing to verify integration of heterologous genes in the targeted locus and complete removal of the gene targeted for deletion from the chromosomal DNA.

**Culture conditions.** Unless otherwise specified, *S. elongatus* cells were cultured in BG11 medium with the addition of 50 mM NaHCO$_3$. Cells were grown at 30 °C with rotary shaking (100 r.p.m.) and light (30 μmol photons · m$^{-2}$ s$^{-1}$ in the PAR range) provided by 86 cm 20 W fluorescent tubes. Light intensity was measured using a PAR quantum flux meter (Model MQ-200, Apogee Instruments). Dark conditions were achieved by wrapping tubes or flasks with aluminium foil and culturing them without light. Cell growth was monitored by measuring OD$_{730}$ in a Microtek Synergy H1 plate reader (BioTek). All OD$_{730}$ values were corrected for 1 cm path length. Cell biomass (dry cell weight (DCW)) was calculated from OD$_{730}$ using the value of 0.22 gDCW l$^{-1}$ per OD$_{730}$ (ref. 44). Antibiotics concentrations were as follows: cycloheximide (50 mg l$^{-1}$), spectinomycin (20 mg l$^{-1}$), kanamycin (20 mg l$^{-1}$), gentamycin (10 mg l$^{-1}$) and chloramphenicol (5 mg l$^{-1}$).

For 23BD production, prior to production experiments, colonies were inoculated in BG11 medium containing 50 mM NaHCO$_3$ and appropriate antibiotics, and grown photoautotrophically. To prepare cells for experiments in continuous dark conditions, pre-grown cells were induced with 0.1 mM IPTG 24 h prior to production tests. Cells at the exponential growth phase were adjusted to an OD$_{730}$ of 5.0 in 10 ml BG11 including 20 mM NaHCO$_3$, 0.1 mM IPTG, 10 mg l$^{-1}$ thiamine and appropriate antibiotics in 20 ml glass tubes with a height of 15 cm and a diameter of 1.5 cm. Appropriate concentration (10 or 15 g l$^{-1}$) of glucose was added as required. Every 24 h, 10% of the culture volume was removed, the pH was adjusted to 7.0 with 3.6 N HCl and volume was replaced with production media containing 200 mM NaHCO$_3$. For 23BD production in diurnal light conditions, 5% of the culture volume was taken every 12 h instead. For long-term production, cells were adjusted to an OD$_{730}$ of 5.0 in 25 ml of BG11 medium in 125 ml baffled glass flasks with a maximum circumference of 33 cm$^2$. The concentration of each medium component was doubled with the exception of HEPES-KOH and A5 trace metals, which remained unchanged and were increased fivefold, respectively. Glucose (15 g l$^{-1}$), NaHCO$_3$ (20 mM), IPTG (1 mM), thiamine (10 mg l$^{-1}$) and appropriate antibiotics were added. On days 3, 6 and 9, cells were collected by centrifugation and resuspended at an OD$_{730}$ of 5.0 in fresh production media

**Quantification of extracellular metabolites.** Glucose concentration in culture supernatant was determined using the D-Glucose Assay Kit (Megazyme Inc.).

For 23BD quantification, culture supernatant samples were analysed using a gas chromatograph (GC; Shimadzu) equipped with a flame ionization detector and HP-5 column (30 m, 0.32 mm internal diameter, 0.25 μm film thickness; Agilent Technologies). The GC oven temperature was increased with a gradient of 40 °C min$^{-1}$ from 70 to 150 °C and held for 2 min. The temperature of the injector and detector was 280 and 330 °C, respectively.

For long-term experiments, accumulative values of 23BD concentration are displayed in Fig. 5a,d. For the first 3 days, values for 23BD correspond to the 23BD concentration in culture supernatants. Following resuspension in fresh production media at 3 days, values correspond to 23BD concentration in culture supernatants in addition to the 23BD concentration at 3 days, prior to resuspension. Values after 6 and 9 days are reported in a similar manner.

Glucose consumption was determined by measuring glucose concentration in culture supernatants at each sampling point and subtracting it from the previous measurement. Glucose concentration was also measured after resuspension in fresh

media. Figure 5b,e corresponds to the accumulative glucose consumption, which is the sum of measured glucose consumption for all prior days.

**GC–MS analysis of $^{13}$C-labelled 23BD.** Prior to analysis, 23BD was derivatized using phenylboronic acid[45]. Briefly, 50 μl of culture supernatant was mixed with 100 μl of acetonitrile containing 100 mg l$^{-1}$ of 1,3-propanediol as an internal standard. A volume of 150 μl of 1,2-dimethoxypropane containing 5 g l$^{-1}$ of phenylboronic acid was added and mixed by vortexing for 10 s. After centrifugation, the organic layer was used for analysis by GC–MS (mass spectrometry). Analysis was performed by GC-8970N (Agilent Technologies) equipped with a VF-5MS column (30 m, 0.25 mm internal diameter, 0.25 μm film thickness; VARIAN) and a GC-5780N mass selective detector (Agilent Technologies) operated at 70 eV. The GC oven temperature was held at 40 °C for 3 min, and then increased at a gradient of 45 °C min$^{-1}$ until 300 °C. The temperature of the injector was set at 225 °C. The ion source (electron ionization) temperature was set at 200 °C. For determination of mass isotopomer distribution, the measured mass spectrum data of the unlabelled 23BD was used to represent any and all combinations of its isotopomer variations[46].

**Quantification of intracellular metabolites.** To prepare samples for metabolomics analysis, cultured cells were collected by vacuum filtration using a nylon membrane filter (0.45 μm, 47 mm, Whatman). Each filter was transferred to 15 ml centrifuge tube, and then immediately frozen in liquid nitrogen and stored at −80 °C until analysis. Metabolite extraction, derivatization and analysis by GC–time of flight–MS was carried out by the West Coast Metabolomics Center at University of California, Davis. Metabolites were identified from MS spectra using the BinBase algorithm[25,26]. For determination of mass isotopomer distribution of alanine and 3-phosphoglycerate, the measured mass spectrum data of the unlabelled material were used to represent any and all combinations of isotopomer variations[46].

Intracellular R15P content was determined enzymatically[22]. Cultured cells (~10 mgDCW) were collected by centrifugation (4,000g, 10 min, 4 °C), frozen in liquid nitrogen immediately and stored at −80 °C until extraction. Intracellular metabolites were extracted with 3 ml of 6% HClO$_4$, and the pH was brought to 7.0 by adding 5 M KOH–1 M triethanolamine solution. Cell extract was obtained by centrifugation (4,000g, 10 min, 4 °C) and subsequently used for R15P measurement. Cell extract was added to the reaction mixture containing 50 mM Tris-HCl (pH 8.0), 10 mM NaHCO$_3$, 1 mM EDTA, 5 mM ATP, 0.3 mM NADH, 5 mM phosphocreatine, 10 mM DTT, 15 mM MgCl$_2$, 2 U creatine kinase, 5 U GAPDH, 8 U 3PGA kinase and 0.1 U RuBisCO. Reaction was performed at 30 °C for 15 min, and the change in absorbance of NADH at 340 nm was monitored to determine R15P content against a standard curve using pure R15P.

**Enzyme assays.** To determine the activities of the following enzymes: G6P dehydrogenase (ZWF), 6PG dehydrogenase (GND), phosphoglucose isomerase (PGI), PRK and RuBisCO, cells were grown with 10 g l$^{-1}$ glucose, 20 mM NaHCO$_3$, 0.1 mM IPTG and 10 mg l$^{-1}$ thiamine for 24 h. Cells were collected, resuspended in 50 mM Tris-HCl buffer (pH 8.0) containing 1 mM DTT and disrupted by a Mini-bead beater to prepare cell lysates. All reactions were performed at 30 °C in reaction mixtures containing 50 mM Tris-HCl buffer (pH 8.0) and 10 mM MgCl$_2$. Enzyme activity was determined by monitoring the change in absorbance of NAD(P)H at 340 nm for 15 min. The reaction mixtures for each enzyme are described as follows: For ZWF, 5 mM G6P and 0.4 mM NADP$^+$ were added. For GND, 5 mM 6PG and 0.4 mM NADP$^+$ were added in a reaction mixture. For PGI assay, 2 mM F6P, 0.4 mM NADP$^+$ and 1 U G6P dehydrogenase were added. For PRK assay, 2 mM Ru5P, 2.5 mM PEP, 2 mM ATP, 0.3 mM NADH, 4 U lactate dehydrogenase and 4 U pyruvate kinase were added. For RuBisCO, 0.5 mM ribulose-1,5-bisphosphate, 10 mM NaHCO$_3$, 1 mM EDTA, 5 mM ATP, 0.3 mM NADH, 5 mM phosphocreatine, 10 mM DTT, 0.4 U creatine kinase, 1 U GAPDH and 1.6 U 3PGA kinase were added.

To prepare cell lysates for assays of 23BD biosynthetic pathway enzymes, cells at exponential phase were diluted to an OD$_{730}$ of 0.1 and cultured in 25 ml of BG11 media containing 50 mM of NaHCO$_3$ in 125 ml shake flasks. Every 24 h the culture pH was adjusted to 7.0 with 3.6 N HCl, and 10% of the media was removed and replaced with fresh media. After 48 h of growth, various concentrations of IPTG were added to the indicated cultures. Cells were collected 24 h after induction by centrifugation, and disrupted using a Mini-bead beater (Biospec Products) to prepare cell lysates. The total protein determination was performed using Advanced Protein Assay Reagent (Cytoskeleton).

To determine the activity of acetolactate synthase, reactions were performed at 30 °C for 15 min in 100 μl of reaction mixtures containing 0.1 M 3-(N-morpholino)propanesulfonic acid (MOPS) (pH 7.0), 1 mM MgCl$_2$, 20 mM pyruvate and 0.1 mM thiamine. By adding 10 μl of 50% H$_2$SO$_4$, reaction was stopped and produced acetolactate was chemically converted to acetoin. A volume of 20 μl of sample was mixed with 480 μl of 0.45 M NaOH, 250 μl of 50 g l$^{-1}$ naphthol and 250 μl of 5 g l$^{-1}$ creatine. Acetoin was quantified by measuring the absorbance at 535 nm against a standard curve using pure acetoin.

To determine the activity of acetolactate decarboxylase, the protocol for the acetolactate synthase assay described above was used with the following

modifications. The substrate was replaced with 2-acetolactate freshly prepared from ethyl-2-acetoxy-2-methylacetoacetate. To prepare 2-acetolactate, 50 µl of ethyl-2-acetoxy-2-methylacetoacetate was mixed with 990 µl of water and 260 µl of 2 M NaOH was gradually added. The acidification step was omitted, and reactions were quenched by the transfer of 20 µl of reaction to wells in a 96-well plate each containing 80 µl 2.5 M NaOH.

**Quantification of green fluorescent protein fluorescence.** In prior to measurements of green fluorescent protein fluorescence, cultured *S. elongatus* cells were collected and resuspended in the equal volume of fresh BG11 medium. For fluorescence measurements, 488 nm was used for excitation, and emission was measured at 530 nm using a Microtek Synergy H1 plate reader (BioTek).

**Data availability.** The authors declare that all data supporting the findings of this study are available within the article, its Supplementary Information file and from the corresponding author upon reasonable request.

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

## Acknowledgements

This work was supported by the Asahi Kasei Corporation and the National Science Foundation (CBET-1132442 and CBET-1349663). A.L.C. is supported by an NSF Graduate Research Fellowship. We thank Nicole E. Nozzi, Anna E. Case and Christine A. Rabinovitch-Deere for critical reading of the manuscript.

## Author contributions

M.K., A.L.C. and S.A. designed research; M.K. and A.L.C. performed the experiments; M.K., A.L.C. and S.A. analysed data; M.K., A.L.C. and S.A. wrote the manuscript.

## Additional information

**Competing financial interests:** The authors declare no competing financial interests.

