## [Peer Review File · Nature Communications]

Editorial Note: This manuscript has been previously reviewed at another journal that is not operating a transparent peer review scheme. This document only contains reviewer comments and rebuttal letters for versions considered at Nature Communications. Mentions of prior referee reports have been redacted.

Reviewer #1 (Remarks to the Author)

I checked the rebuttal letter and the revised manuscript. I believe that authors have addressed the comments and improved the manuscript. The current version is suitable for publication in Nature Communication.

Reviewer #2 (Remarks to the Author)

The authors have adequately addressed my concerns. I still have my doubts about the advantage of mixotrophic production over purely heterotrophic or photoautotrophic system. However, it does not compromise the achievements of the study.

Reviewer #3 (Remarks to the Author)

I reviewed the earlier version of this manuscript, and I am happy to see the changes that have been made. In particular, I appreciate the increased attention to the step-by-step enunciation of the logic that went into the experimental design. This is a well-developed story about cyanobacterial biotechnology that will be appreciated by the readers.

Minor comments:

I remain a bit confused about the answer regarding biosynthetic intermediates decreasing despite growth increasing. The answer was that there was unlikely to have been steady-state growth. Over the timeline of hours, that may have been true, but metabolism is so fast that it will readily attain a quasi-steady state for which concepts like metabolic control analysis will still apply. Given that will be the case, I think the findings are interpretable from the point of view presented before:

“This may at first seem counterintuitive, but all metabolites are at steady-state if exponential growth is maintained, whereby flux in and out are necessarily balanced. Thus it is actually quite common for currency metabolites (like NADPH) to be lower when growth is faster because they are being used more quickly and the new steady state is actually a lower concentration.”

If you want to read about other scenarios where this has been seen, we have come across it in our own work (Carroll and Marx, 2013. PLoS Genetics) and we cited a couple other examples (Kayser et al., 2005. Microbiology; Lahtvee et al., 2011. Microb Cell Fact).

We appreciate all of the comments we have received from the reviewers.

REVIEWERS' COMMENTS:

Reviewer #1 (Remarks to the Author):

I checked the rebuttal letter and the revised manuscript. I believe that authors have addressed the comments and improved the manuscript. The current version is suitable for publication in Nature Communication.

--- Thank you for your encouraging remarks.

Reviewer #2 (Remarks to the Author):

The authors have adequately addressed my concerns. I still have my doubts about the advantage of mixotrophic production over purely heterotrophic or photoautotrophic system. However, it does not compromise the achievements of the study.

--- We appreciate the positive comments. We hope that we will be able to convince this reviewer about the advantages of mixotrophic production over purely heterotrophic or photoautotrophic systems with the future publications.

Reviewer #3 (Remarks to the Author):

I reviewed the earlier version of this manuscript, and I am happy to see the changes that have been made. In particular, I appreciate the increased attention to the step-by-step enunciation of the logic that went into the experimental design. This is a well-developed story about cyanobacterial biotechnology that will be appreciated by the readers.

Minor comments:

I remain a bit confused about the answer regarding biosynthetic intermediates decreasing despite growth increasing. The answer was that there was unlikely to have been steady-state growth. Over the timeline of hours, that may have been true, but metabolism is so fast that it will readily attain a quasi-steady state for which concepts like metabolic control analysis will still apply. Given that will be the case, I think the findings are interpretable from the point of view presented before:

“This may at first seem counterintuitive, but all metabolites are at steady-state if exponential growth is maintained, whereby flux in and out are necessarily balanced. Thus it is actually quite common for currency metabolites (like NADPH) to be lower when growth is faster because they are being used more quickly and the new steady state is actually a lower concentration.”

If you want to read about other scenarios where this has been seen, we have come across it in our own work (Carroll and Marx, 2013. PLoS Genetics) and we cited a couple other examples (Kayser et al., 2005. Microbiology; Lahtvee et al., 2011. Microb Cell Fact).

--- Thank you for your helpful comments for improving the manuscript. We have added the references suggested by this reviewer and modified the text include this explanation. It now reads: “Another possibility is that these metabolites have a high in/out flux as a result of rapid growth, and are at a lower steady-state concentration. This has been observed in flux-analysis of fast-growing cultures³⁴⁻³⁶. However, the lower concentrations of these metabolites in **Strain 15** compared to **Strain 11** did not correlated with a faster growth rate, only an increase in 23BD production (**Fig. 4a & b**).” (Page 14 Line 10).